# Quality and Price of Spruce Logs, Determined Conventionally and by Dendrochronological and NDE Techniques

Aleš Straže [ID], Klemen Novak and Katarina Čufar *[ID]

Biotechnical Faculty, University of Ljubljana, Jamnikarjeva 101, 1000 Ljubljana, Slovenia;
ales.straze@bf.uni-lj.si (A.S.); klemen.novak@bf.uni-lj.si (K.N.)
* Correspondence: katarina.cufar@bf.uni-lj.si; Tel.: +386-1-320-3645

**Abstract:** We examined valuable log assortments of Norway spruce (*Picea abies*) from a traditional auction in Slovenia where spruce growth on many sites is affected by climate change. From 6620 logs, we selected 817 that obtained the highest prices. Factors including log dimensions and geometry, tree-ring characteristics, quality grades according to the standard, properties measured by NDE stress wave testing, and their combined effect on price were modelled. The results showed that half of the auctioned logs were of highest quality ($Q_1$, $Q_2$), with diameters over 60 cm. These logs were more expensive than the thinner logs of lower quality ($Q_3$, $Q_4$). The quality class of the logs, determined by their external features and geometry, was associated with tree-ring and acoustic characteristics. The artificial neural network model (ANN) with feed-forward backpropagation using tree-ring data, longitudinal stress wave velocity, and damping showed that more than 75% of the logs could be accurately classified into quality classes. On the other hand, tree-ring data and acoustic characteristics could not adequately explain the price offered at auction, which probably also depends on unidentified individual requirements and the needs of the buyer.

**Keywords:** Norway spruce (*Picea abies*); log quality; tree-ring characteristics; NDE measurements; price; auction



## 1. Introduction

Norway spruce (*Picea abies* Karst.) is an important tree species in Europe, providing a widely used industrial wood used for a variety of products, such as pulp and paper, various composites, panels, interior and exterior structures, furniture, and many other items [1]. High-quality spruce wood is also essential for the production of veneer and special products such as musical instruments. The quality requirements for musical instruments are very high, although the definition of quality is still incomplete, affected by both needs and markets [2–9]. The issue of spruce wood quality has therefore attracted the interest of forest owners and managers, wood technologists, timber merchants, musical instrument manufacturers, and others.

Norway spruce is the second most common tree species in Slovenian forests and accounts for 30% of the wood stock, with a volume of over 100,000,000 m$^3$ [10]. Native to the montane areas of the Alps and the Dinaric Mountains, spruce has been planted on a large scale since the 19th century and thus has spread throughout entire country [11], even to lowland sites outside its natural range. Therefore, it currently grows on numerous sites, where it is affected by climate change and associated negative impacts such as storms, heat waves, and insect infestations, causing its retreat in many locations. In the future, this is expected to affect wood quality and availability of high-quality wood assortments.

After tree felling and crosscutting, the quality of sawlogs is, as a rule, assessed according to national regulations (Table 1), which like other European standards define the required size and geometry of logs and the permissible size and quantity of visible anomalies [12]. The quality of logs typically declines from the base of the trees to the

canopy, although the factors affecting this are complex [7,13,14]. From the point of view of structural quality and geometric characteristics, butt logs from trees with appropriate social status and from a suitable habitat are particularly relevant, as they can fetch very high prices on the market.

**Table 1.** Characteristics and threshold values for classifying logs into five quality classes according to the Slovenian regulations for the classification of forest timber assortments from state forests ($Q_1$, $Q_2$—exceptional, veneer quality logs; $Q_3$–$Q_5$—logs for sawn timber of excellent, medium, and poor quality; N.A.—not allowed).

| Characteristics | | Quality Class | | | | |
|---|---|---|---|---|---|---|
| | | $Q_1$ | $Q_2$ | $Q_3$ | $Q_4$ | $Q_5$ |
| Dimensions of log | Mean diameter (cm) | >45 | >40 | > 35 < 55 | >20 | >20 |
| | Length (m) | >3 | >3 | >3 | >3 | >3 |
| Knots | Sound | N.A. | N.A. | N.A. | $D \leq 4$ cm | $D \leq 8$ cm |
| | Dead | N.A. | N.A. | N.A. | N.A. | $D \leq 4$ cm |
| Eccentricity [%] | | N.A. | N.A. | $\leq 10$ | $\leq 15$ | unlimited |
| Sweep [cm/m] | $20 \leq D \leq 35$ cm | N.A. | N.A. | - | $\leq 1.0$ | $\leq 2.0$ |
| | $D \geq 35$ cm | | | $\leq 1.0$ | $\leq 1.5$ | $\leq 2.0$ |
| Taper | Length $\leq 6$ m — $20 \leq D \leq 35$ cm | N.A. | N.A. | - | $\leq 1.2$ | $\leq 1.7$ |
| | $D \geq 35$ cm | | | - | $\leq 1.7$ | $\leq 2.6$ |
| | Length > 6 m — $20 \leq D \leq 35$ cm | - | - | - | $\leq 1.1$ | $\leq 1.4$ |
| | $D \geq 35$ cm | | | - | $\leq 1.3$ | $\leq 1.6$ |
| Heart cracks | | N.A. | N.A. | $\leq D/4$ | $\leq D/3$ | $\leq D/2$ |
| Ring shakes | | N.A. | N.A. | N.A. | $\leq D/4$ | $\leq D/3$ |

Most of the spruce in Slovenia grows in small (average 2.9 ha) private forest stands [10]. Therefore, small owners welcome the opportunity to sell the high-value timber at a central timber auction held every year in Slovenj Gradec, Slovenia (46.49° N, 15.07° E). This auction of the most valuable wood has been organized annually since 2007 by the Association of Forest Owners of Mislinjska dolina and the Association of Slovenian Forest Owners, with the support of the Slovenian Forest Service. It enables the forest owners to sell top quality timber at higher prices than through regular timber purchases. At the auction, buying and selling consists of offering the wood for bids, taking the bids of potential buyers, and then selling the wood to the highest bidder. The logs are on a show before the auction, so that the potential buyers can inspect the assortments and submit closed bids to the timber auction contractor. Every log is then sold to the highest bidder at the end of the auction. Finally, the prices obtained for the logs are published along with information about each log, including the dimensions. Some basic data on the origin, owner, and buyer of each log are collected as well.

The objective of our study was to evaluate the quality parameters of the most valued Norway spruce logs using stress wave measuring, tree-ring characteristics, log external features, and geometry data to determine how these parameters and their combinations predict the quality rating and the price obtained for logs at auction. Finally, artificial neural network design modelling was used to propose a log grading system based on stress wave tests and tree-ring characteristics.

## 2. Materials and Methods

### 2.1. Log Sampling

The study included valuable log assortments of Norway spruce (*Picea abies* Karst.) from the 14th auction in Slovenj Gradec 2020, Slovenia. In total, 6620 logs from 33 different tree species were offered, with a total volume of 6614 m³, of which 846 were Norway spruce.

For our analysis, we selected the 817 highest priced logs, with a total volume of 1441 m$^3$, for which 3642 bids were received at the auction (Figure 1).

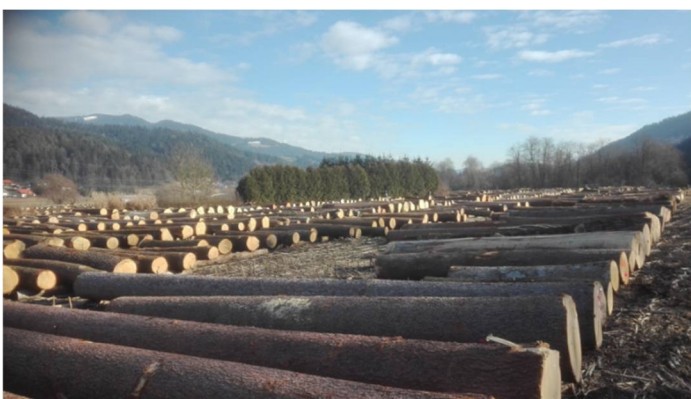

**Figure 1.** Valuable spruce logs and logs of other wood species on display at the auction field.

Before the auction, interested buyers received the assortment catalogue with the tree species and the diameter and length of the logs, but without any information on quality assessment. After a visual inspection of the logs, closed bids were submitted to the timber auction contractor. The bidder sold a single log to the highest bidder at the end of the auction. We conducted our sampling after the assortments had already been sold, and therefore knew the prices. We selected the logs for analysis based on the prices obtained.

Using the auction organizer's catalogue, we divided the logs into diameter classes. The diameter of the logs, determined at the mid-section of the log, ranged from 43 to 84 cm, which was the basis for dividing the logs into diameter classes (n = 9) with a 5 cm interval: $D_1$ (<45 cm), $D_2$ (45–49 cm), $D_3$ (50–54 cm), $D_4$ (55–59 cm), $D_5$ (60–64 cm), $D_6$ (65–69 cm), $D_7$ (70–74 cm), $D_8$ (75–79 cm), and $D_9$ (80–84 cm). Since the $D_1$ class was not sufficiently represented, we did not include it in the analysis.

All the assortments offered at the auction corresponded to the butt log of the tree. The length of the logs ranged from 3.0 to 11.0 m ($L_L$), and was used to sort logs into three length classes: $L_1$ ($\leq$4 m), $L_2$ (4–8 m), and $L_3$ (8–12 m) (Figure 2).

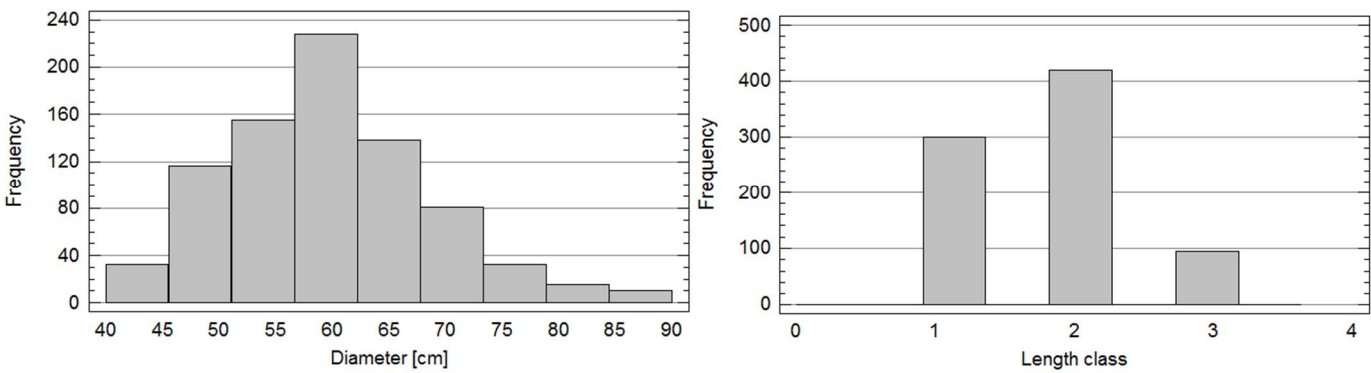

**Figure 2.** Frequency distribution of 817 high-priced spruce logs by diameter (**left**) and length classes (**right**).

### 2.2. Visual Determination of Log Quality According to Standards

The auction organizer classified the logs in accordance with Slovenian regulations for classification of forest timber assortments from state forests, which arrange the quality into five classes: F, L, A, B, and C, renamed $Q_1$–$Q_5$ in this study. Logs of exceptional quality ($Q_1$ and $Q_2$) have a diameter greater than 45 or 40 centimetres, respectively, and must be free of anomalies, such as knots, sweep, taper, eccentricity, end cracks, and discoloration. The width of the annual growth rings must be uniform and less than 6 mm. Characteristics

of $Q_1$ are adequate for sliced veneer; those of $Q_2$ are adequate for rotary cut veneer production (Table 1).

The classes $Q_3$, $Q_4$, and $Q_5$ are classified for the production of excellent ($Q_3$), medium ($Q_4$), and poor quality ($Q_5$) sawn timber. In these sawlogs, the presence of structural and other anomalies is allowed, the size and frequency of which increases with decreasing quality class (Table 1).

### 2.3. Determination of Stress Wave Velocity and Vibration Damping in Logs by Vibration Resonance Method

Longitudinal vibration was measured on a subset of 59 logs to evaluate the mechanical properties, and thus additionally the quality of the wood. These logs mainly originated from mixed forest sites in the surroundings of Slovenj Gradec, Slovenia (46.52° N, 15.07° E) and Ribnica (45.77° N, 14.73° E) at elevations below 1000 m a. s. l.

For the longitudinal vibration, the logs were placed on round wooden supports placed near both ends of the logs. The logs were excited from the front end with a steel hammer with a mass of 500 g, where at the same location, the sound signal was recorded using a unidirectional condenser microphone (PCB-130D20; PCB Piezotronics Inc., Depew, NY, USA) (Figure 3a). The signal was acquired using an NI-9234 DAQ-module (National Instruments Inc., Austin, TX, USA) in 24-bit resolution with a 51.2 kHz sampling frequency. The measured natural frequency of the log ($f_L$) in the first vibration mode was used to determine the stress wave velocity (*SWV*) in the longitudinal direction of the log (*SWV*; Equation (1); Figure 3b). This fundamental wave equation was developed for idealized elastic materials in the form of a long slender rod of length *L* [15,16].

$$SWV = 2f_L L \qquad (1)$$

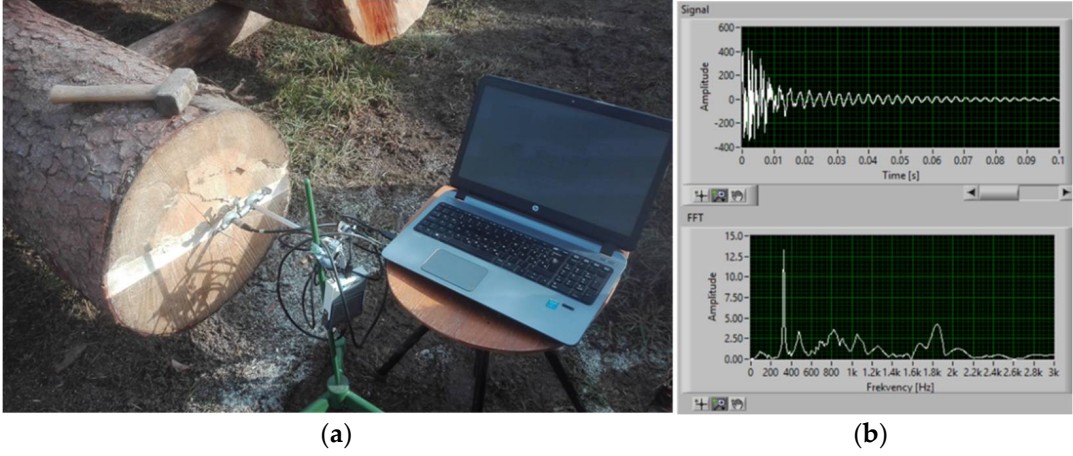

**Figure 3.** Acquisition of the sound signal at longitudinal vibration of logs (**a**) and recorded sound scheme (**b**), using least-squares regression analysis (*α*—temporal damping), whereby the value of the damping coefficient, i.e., loss tangent (tan *δ*) was determined (Equation (2)).

$$\tan \delta = \frac{\alpha}{\pi \cdot f_L} \qquad (2)$$

### 2.4. Dendrochronological Measurements and Analysis

The dendrochronological analysis was performed on the same subset of 59 logs as the acoustic measurements and on a cross-section of each log, which corresponded to the height level in a standing tree of 4–4.5 m above the ground. Since the usual destructive sampling by boring was not possible, we prepared a smooth surface along two radii of the log and took images.

For smoothing, a hand-held cordless trimmer (Makita DRT 50ZX2, Anjo, Japan) was used to cut a groove about 25 mm wide and 5 mm deep in the cross section, from the bark over the pith to the bark on the other side. To ensure an even cut, a wooden support rail was attached to the log end, along which the trimmer was guided. The sawdust was then removed to make the growth ring boundaries visible (Figure 4a). Scale bars were attached to calibrate the images. A digital camera (Canon EOS70D, Tokyo, Japan) with tripod mount was used to photograph the radii (Figure 4b). The photos were later processed in Adobe Photoshop Elements 2020 and prepared for measurements.

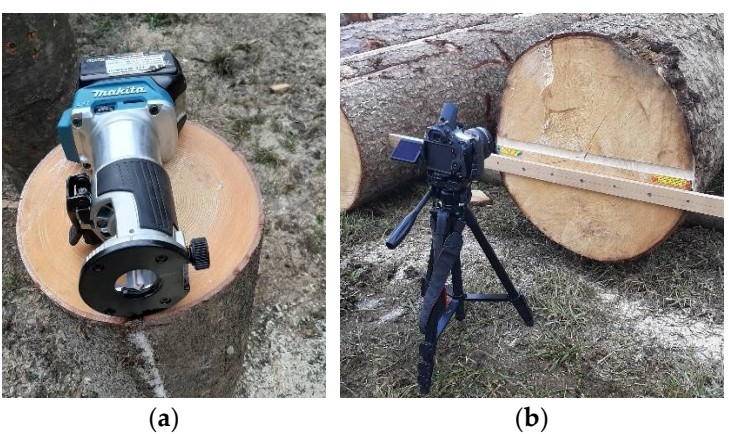

(**a**)      (**b**)

**Figure 4.** The principle of creating a groove in the cross-section of the log (**a**) and the image acquisition (**b**).

The CooRecorder 9.6 (Cybis Elektronik & Data AB) program was used for tree-ring, earlywood, and latewood width measurements (Figure 5). Tree-ring series were opened in the CDendro portion of the program for verification and conversion to *fh or *rwl format. Crossdating was performed using the TSAP Win program ® software (Rinntech, Heidelberg, Germany), and the data were processed and exported for further analysis.

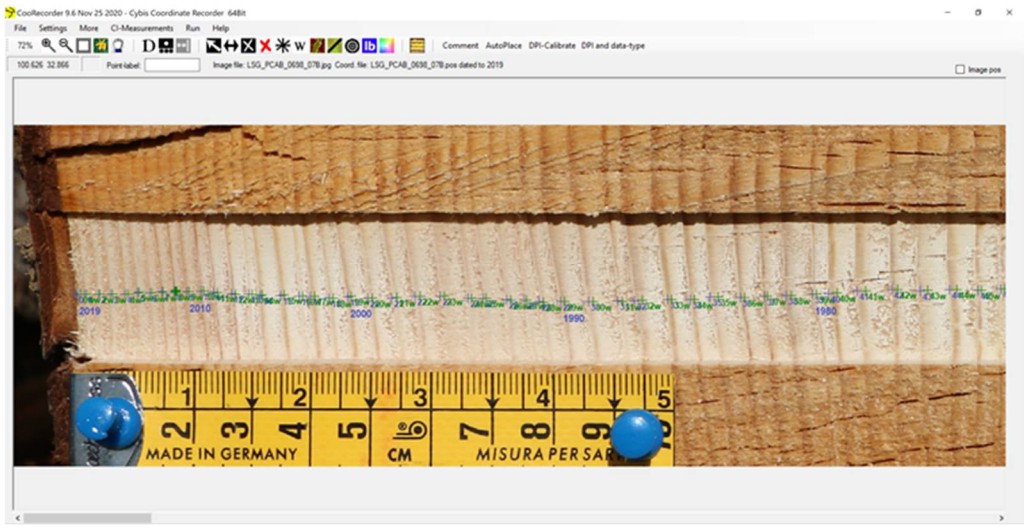

**Figure 5.** Tree-ring, earlywood, and latewood width measurements in CooRecorder on the image of a groove smoothed with a hand-held cordless trimmer.

### 2.5. Data Analysis and Modelling

In addition to basic data analyses in CooRecorder, TSAP Win, and MS Excel, statistical analysis was also performed in Statgraphics Centurion 15.2. (Statgraphics Technologies, The Plains, VA, USA). We used ANOVA ($p < 0.05$) to examine individual variables such as log price, diameter (*D*), eccentricity (*ECC*), length (*L*), stress wave velocity (*SWV*), vibration damping (tan *δ*), tree-ring width (*RW*), and proportion of latewood (*LWP*). Interdependence

and influence were also analysed using a multiple range test (Duncan, $p < 0.05$) and multifactorial analysis (MANOVA, $p < 0.05$). In addition, linear regression analysis was used to analyse the correlations among variables and determine radial growth profiles.

Classification of Logs by Quality and Price with an Artificial Neural Network

An artificial neural network (ANN) [17] using a feed-forward backpropagation was chosen to model the individual variables of every log ($D$, $ECC$, $L$, $SWV$, tan $\delta$, $RW$, $LWP$) to identify a set of features for classification. To improve the operation of the transfer functions ($f$), the input and output data were normalized using Equation (3), which made the transfer function more effective by producing an output in the interval ($-1$, $1$), and improved the network's ability to generalize [17,18].

$$X' = \frac{X - X_{min}}{X_{max} - X_{min}} \tag{3}$$

where $X'$ is the value after normalization of the vector $X$, and $X_{min}$ and $X_{max}$ are the minimum and maximum values of the vector $X$. The hyperbolic tangent was chosen for the transfer function ($f$) ($f(X) = \tanh(X)$), where $f(X)$ is the output value of the neuron and $X$ is the input value of the neuron. Based on previous research [19–21], we used a single hidden layer of 12 neurons, which was found to have effective speed of convergence during training, and on the accuracy of classification. We investigated the possibility of classifying logs into four quality classes ($Q_1$, $Q_2$, $Q_3$, $Q_4$; there was no log from quality class $Q_5$) and into nine price classes ($P_1$–$P_9$) defined with a range of 100 EUR/m$^3$. The lowest price class was $P_1$ ($<$100 EUR/m$^3$) and the highest was $P_9$ ($800 < P_9 < 900$ EUR/m$^3$). The number of neurons in the output layer corresponded to the number of classes, with four neurons outputting results for the classification of logs by quality (Figure 6a) and nine neurons outputting results for the classification by price (Figure 6b).

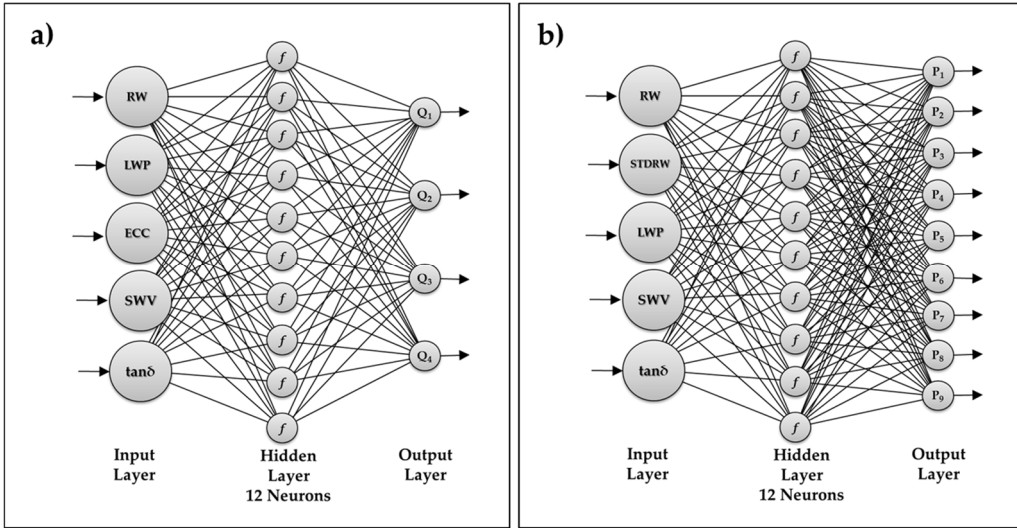

**Figure 6.** Topology of the artificial neural network built with an optimized set of input variables (*RW*—mean ring width, *STDRW*—standard deviation of ring widths in individual log, *LWP*—latewood percentage, *ECC*—log eccentricity, *SWV*—stress wave velocity, tan $\delta$—vibration damping) to classify logs into quality (**a**) and price classes (**b**). $Q_1$–$Q_4$—quality classes; $P_1$–$P_9$—price classes (lowest–highest).

Seventy percent of the data was used to train the network; the rest was used to test the accuracy of the model. Network error was calculated by comparing the model output to the target value in Statgraphics Centurion 15.2 software. Backpropagation with a gradient descent optimization method was used to adjust the weights of the neurons in the training process. The performance of the ANN was evaluated using the confusion matrix to compare the predicted class with the actual class (Figures 6 and 7).



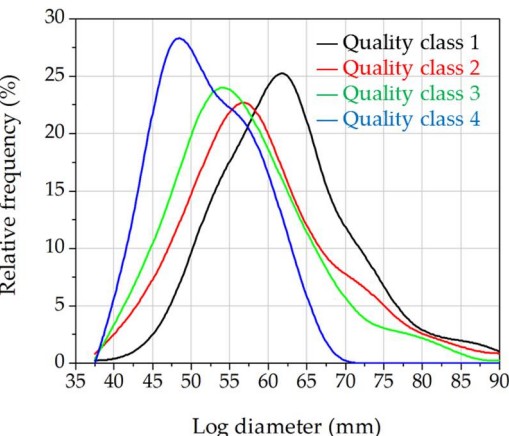

**Figure 7.** Frequency distribution of diameter of spruce logs from various quality classes ($Q_1$ and $Q_2$—veneer log quality, $Q_3$ and $Q_4$—sawlog quality).

## 3. Results and Discussion

### 3.1. Interdependence between Quality According to Standards, Geometric Characteristics of Logs, and Tree-Ring Characteristics

The spruce logs of the $Q_1$ quality class were on average the thickest, and as the quality of the logs decreased, their mean diameter also gradually decreased ($D_{Q1}$ = 61.8 cm, St.dev = 8.56; $D_{Q2}$ = 58.9 cm, St.dev = 9.02; $D_{Q3}$ = 56.1 cm, St.dev = 9.67; $D_{Q4}$ = 55.6 cm, St.dev = 7.06; St.dev—standard deviation) (Figure 7). The Duncan multiple range test ($p < 0.01$) only revealed a significant difference in diameter between $Q_1$ and $Q_3$-$Q_4$ logs. The reduction in the number of defects and, conversely, the increase in quality with the increasing diameter of spruce logs is in agreement with other studies [6], and is also influenced by forest quality [3] and site stand density [22].

On average, the logs contained 113 (range 74–167) tree rings at a tree level equivalent to 4 m above ground in the tree. As expected, higher quality logs, which were also thicker, had more growth rings (Table 2). On average, the estimated age of trees was around 130 years, i.e., 15–20 more than the number of tree rings at the level of 4 m.

**Table 2.** Variation in number of tree rings in spruce logs of quality classes from $Q_1$ (highest) to $Q_4$ (lowest). St.dev—standard deviation; CoV—Coefficient of variation.

| Quality Class | Number of Logs | Number of Tree Rings | | | | |
|---|---|---|---|---|---|---|
| | | Mean | St.dev | CoV (%) | Minimum | Maximum |
| $Q_1$ | 26 | 121 | 19.5 | 15.7% | 96 | 167 |
| $Q_2$ | 13 | 118 | 17.0 | 10.1% | 90 | 153 |
| $Q_3$ | 17 | 111 | 12.6 | 17.6% | 79 | 145 |
| $Q_4$ | 3 | 104 | 12.0 | 11.6% | 74 | 138 |
| Total | 59 | 113 | 15.3 | 13.7% | 74 | 167 |

The average tree-ring width (*RW*) of the examined logs was 2.65 mm (CoV = 23.18%). The average ring width in the studied logs increased with increasing diameter (ANOVA, $p < 0.01$). A statistically significant greater *RW* was observed in logs of higher diameter classes ($D_6$–$D_9$) compared to the smaller diameter classes ($D_1$–$D_4$) ($RW_{D2}$ = 2.12 mm, CoV = 35.35%; $RW_{D3}$ = 2.05 mm, CoV = 13.08%; $RW_{D4}$ = 2.29 mm, CoV = 14.41%; $RW_{D5}$ = 2.62 mm, CoV = 16.78%; $RW_{D6}$ = 2.90 mm, CoV = 11.76%; $RW_{D7}$ = 2.86 mm, CoV = 11.38%; $RW_{D8}$ = 2.80 mm, CoV = 24.97%; $RW_{D9}$ = 3.83 mm, CoV = 18.67%; Figure 8a). On average, wider growth rings, but without statistical significance (ANOVA, $p = 0.11$), were also found in higher quality logs, where the variation was also somewhat higher ($RW_{Q1}$ = 2.85 mm, CoV = 25.81%; $RW_{Q2}$ = 2.36 mm, CoV = 23.65%; $RW_{Q3}$ = 2.62 mm, CoV = 16.62%; $RW_{Q4}$ = 2.55 mm, CoV = 17.02%) (Figure 8b).

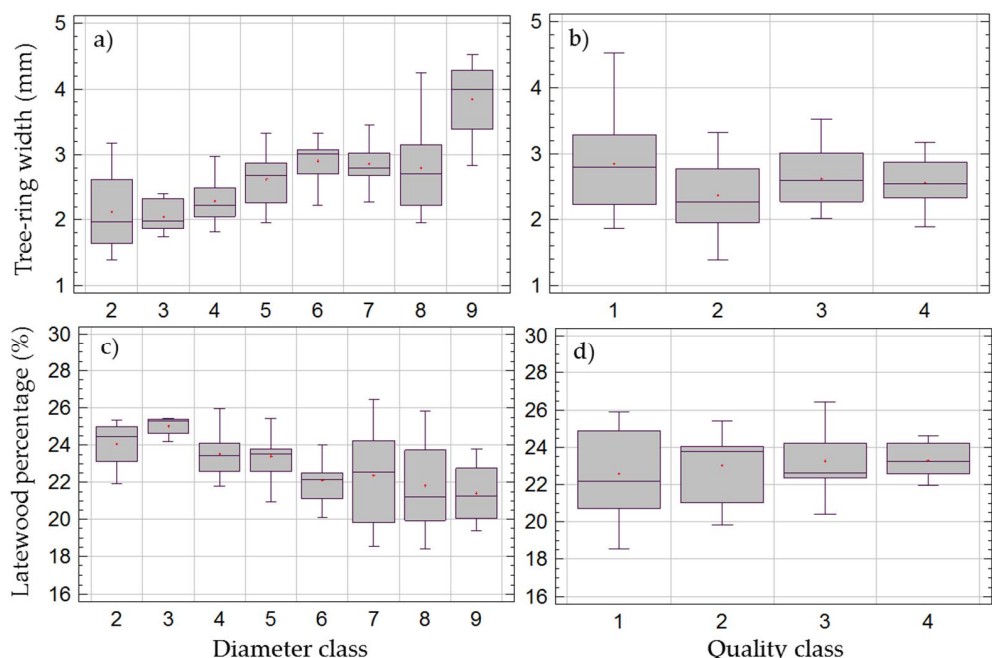

**Figure 8.** Distribution and interdependence between tree-ring width in relation to the log diameter (**a**) and quality class (**b**); latewood percentage in relation to the log diameter (**c**) and quality class of logs (**d**).

The average percentage of latewood (*LWP*) in the studied logs was 22.94% (CoV = 8.30%). The proportion of latewood in the studied logs decreased only slightly with increasing log diameter. However, the differences between log diameter classes were not statistically significant (ANOVA, *p* = 0.06), primarily due to greater variability in thicker assortments ($LWP_{D2}$ = 24.06%, CoV = 6.13%; $LWP_{D3}$ = 25.03%, CoV = 1.99%; $LWP_{D4}$ = 23.53%, CoV = 4.88%; $LWP_{D5}$ = 23.40%, CoV = 5.68%; $LWP_{D6}$ = 22.10%, CoV = 5.72%; $LWP_{D7}$ = 22.38%, CoV = 11.22%; $LWP_{D8}$ = 22.03%, CoV = 10.16%; $LWP_{D9}$ = 21.41%, CoV = 8.67%). The differences in the proportion of latewood were not significant with regard to the quality classes ($LWP_{Q1}$ = 22.58%, CoV = 9.99%; $LWP_{Q2}$ = 23.03%, CoV = 8.62%; $LWP_{Q3}$ = 23.25%, CoV = 7.00%; $LWP_{Q4}$ = 23.29%, CoV = 4.13%; Figure 8c,d).

The relationship between ring width and the proportion of latewood showed a weak negative trend (Figure 9). For the ring width–latewood proportion/wood density relationship, the literature also mostly reports inverse but weak relationships [5,23,24]. The weak correlation is probably due to the significant influence of other characteristic factors, such as age-related changes in the proportion of latewood [25], climatic parameters [26–28], and social status of forest trees [22], which all affect latewood formation.

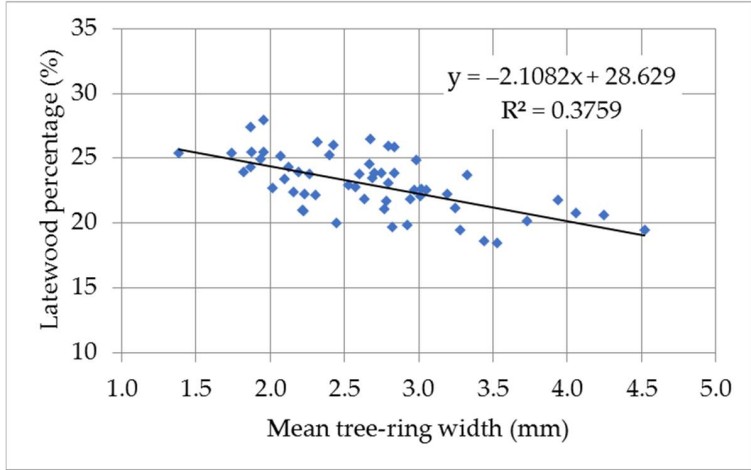

**Figure 9.** Relationship between latewood percentage and mean ring width of tested spruce logs.

### 3.2. Log Characteristics and Quality Evaluated by Stress Wave Velocity

The average stress wave velocity in the tested spruce logs was 3968 m/s (CoV = 4.92%). Thinner logs ($D_2$–$D_6$) had a slightly higher average stress wave velocity of about 4000 m/s. Thicker logs ($D_7$–$D_9$) had an average stress wave velocity slightly lower than 3900 m/s. Otherwise, no statistically significant differences in the speed of the stress waves were found with respect to the diameter of the logs (ANOVA, $p = 0.11$) (Figure 10a).

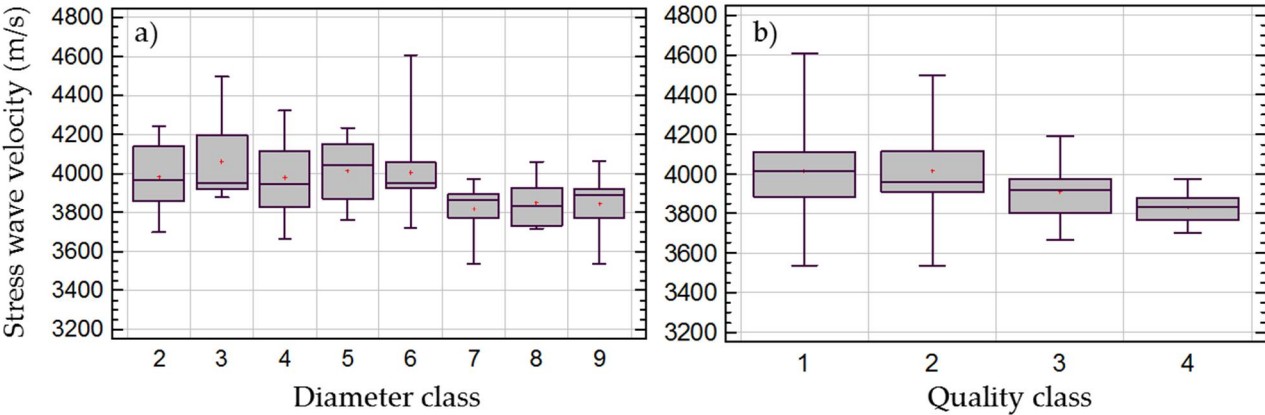

**Figure 10.** Distribution and interdependence between stress wave velocity and (**a**) diameter class, and (**b**) quality class of spruce logs.

However, we confirmed significant dependence of the stress wave velocity on the quality class of the logs. In this case, the decrease in the log quality class contributed to the decrease in the speed of stress waves ($v_{Q1} = 4016$ m/s, CoV = 5.50%; $v_{Q2} = 4021$ m/s, CoV = 5.11%; $v_{Q3} = 3915$ m/s, CoV = 3.87%; $v_{Q4} = 3834$ m/s, CoV = 3.48%) (Figure 10b).

If only clear wood is present, we might also expect a higher stress wave velocity in low-grade logs because they are generally thinner and, on average, have narrower ring widths and a higher proportion of latewood, which positively affect the density and mechanical properties of the wood [29–31]. The expected inverse relationship between stress wave velocity and ring width and positive relationship between stress wave velocity and latewood percentage were found in the tested spruce logs (Figure 11). Both relationships are weak, and the data scatter could be due to other factors affecting longitudinal stress wave velocity in conifers. Recent studies on the quality of large-diameter round spruce wood confirmed that density increases and the knot area ratio decreases with increasing distance from the pith [32,33]. These two characteristics have a strong influence on wood quality. The knot area ratio, along with fibre angle, presence of reaction wood, microfibril angle, and fibre length, are the most influential factors in stress wave propagation in standing trees and logs [4,34–36]. Other similar studies have also confirmed the negative influence of the geometric shape of the logs, such as an irregular shape in cross section, sweep, taper, and other growth defects on stress wave velocity, and that dynamic mechanical strength and stiffness are more common in low-grade logs [37]. Finally, it should be mentioned that the stress waves were analysed in the green state of the logs, where their velocity is not influenced by the moisture content [38,39], but rather by the factors mentioned above.

The average vibration damping (tan $\delta$) of spruce logs was 0.026 (CoV = 19.39%). The average value was high, due to the green state of the wood (*MC* > *FSP*; *MC*—moisture content, *FSP*—fibre saturation point). No statistically significant difference in the vibration damping was found with respect to the diameter of the logs (ANOVA, $p = 0.20$). Several low values of vibration damping were found in $Q_1$-grade logs (tan $\delta_{min} = 0.013$), with the highest values for $Q_3$ and $Q_4$ logs (tan $\delta_{max} = 0.036$). Average vibration dumping was significantly higher in $Q_4$ (tan $\delta_{Q4} = 0.028$) than in $Q_1$ and $Q_2$ logs (tan $\delta_{Q1,Q2} = 0.024$) (ANOVA, $p = 0.04$) (Figure 12).

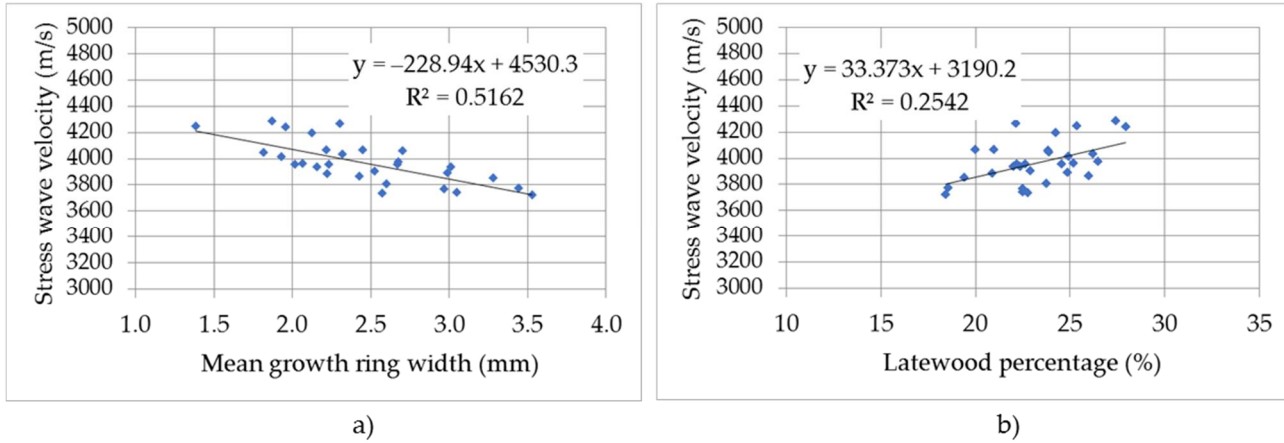

**Figure 11.** Relationship between longitudinal stress wave velocity and (**a**) growth ring width, and (**b**) the latewood percentage.

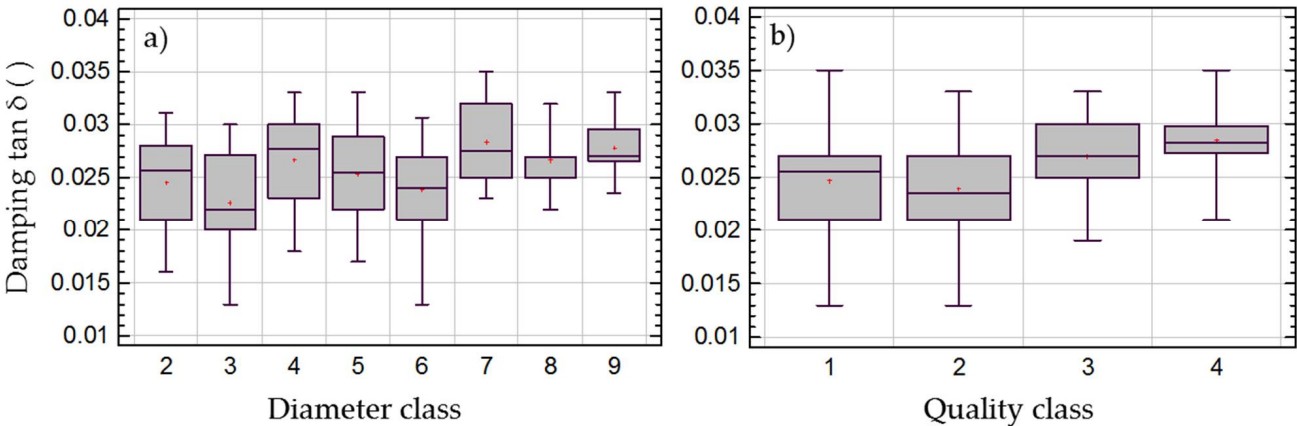

**Figure 12.** Distribution and interdependence between vibration damping (tan $\delta$) and the (**a**) diameter class of logs and (**b**) quality class ($Q_1$–$Q_4$) of spruce logs.

According to these results, damping under cyclic stress is caused by internal friction throughout the material structure (material damping) and by energy dissipation associated with junctions or interfaces between parts of the structure (structural damping) [40,41]. The latter may explain the observation that vibration damping is greatest in $Q_4$ logs, with numerous internal defects which are larger in size and frequency and cause greater energy dissipation than in higher quality grades of wood. To date, damping of mechanical vibration has not been directly studied in relation to log or structural lumber quality. It has been found that the increase in vibration damping is significant in logs subjected to decay [42,43], and constant when lumber moisture content (*MC*) is above the fibre saturation point [44].

### 3.3. Relationship between the Technological Characteristics of Logs and the Price

The prices offered at the auction ranged from 44 to 3600 EUR per log, with an average of 382 EUR (CoV = 111.4%), and were related to the volume of the logs, which ranged from 0.45 to 7.04 m$^3$ (CoV = 47.2%). The price recalculated per 1 m$^3$ ranged from 70.5 to 890 EUR/m$^3$ (CoV = 74.1%).

In the multifactor analysis (MANOVA, $p < 0.05$) where diameter, quality, and length classes were tested as factor variables, only the diameter and quality classes showed a significant impact on the price (Table 3).

**Table 3.** Significance of the effect of diameter, quality, and length classes on the price obtained per cubic metre for the spruce logs at the auction (MANOVA, $p < 0.05$).

| Source | Sum of Squares | Df | Mean Square | *F*-Ratio | *p*-Value |
|---|---|---|---|---|---|
| MAIN EFFECTS | | | | | |
| A: Diameter class | $1.415 \cdot 10^6$ | 8 | 176,905.0 | 8.96 | 0.000 |
| B: Quality class | 159,832.0 | 4 | 53,277.4 | 2.70 | 0.045 |
| C: Length class | 99,810.7 | 3 | 33,270.2 | 1.68 | 0.169 |
| RESIDUAL | $1.584 \cdot 10^7$ | 802 | 19,753.0 | | |
| TOTAL (CORRECTED) | $1.787 \cdot 10^7$ | 816 | | | |

As expected, using the multiple range test (Duncan, $p < 0.05$), we confirmed the positive influence of log diameter on the price (Table 4). This is partly due to the larger diameters of high-quality logs (Figure 7). The higher price obtained for thicker logs can also be attributed to the expected higher efficiency and material yield during processing in a sawmill or veneer production plant [2,45]. The frequency distribution of the logs in diameter classes 6–9 shows that the proportion of $Q_1$ and $Q_2$ logs is 85%, where 65% are $Q_1$ logs and 20% are $Q_2$ logs. Thus, the proportion of sawn logs with a diameter greater than 65 cm ($>D_6$) is only 15%, which is another reason why higher prices are obtained for thicker logs.

**Table 4.** Dependence of the price per cubic metre and diameter class of the spruce logs (Duncan, $p < 0.05$).

| Diameter Class | Diameter (cm) | n | Price per Unit Volume (EUR/m³) | Variance (EUR/m³) | Homogeneous Groups | Significantly Different from Class |
|---|---|---|---|---|---|---|
| 2 | 45–49 | 73 | 113.3 | 32.1 | X | 5, 6, 7, 8, 9 |
| 3 | 50–54 | 157 | 135.7 | 30.1 | XX | 6, 7, 8, 9 |
| 4 | 55–59 | 181 | 159.6 | 29.7 | XX | 6, 7, 8, 9 |
| 5 | 60–64 | 189 | 180.7 | 29.4 | XX | 8, 9 |
| 6 | 65–69 | 93 | 218.1 | 31.7 | X | 1, 2, 8, 9 |
| 7 | 70–74 | 65 | 219.4 | 33.5 | X | 1, 2, 8, 9 |
| 8 | 75–79 | 32 | 277.7 | 38.5 | X | 1, 2, 3, 4, 5, 6, 7 |
| 9 | >80 | 27 | 280.3 | 37.5 | X | 1, 2, 3, 4, 5, 6, 7 |

If we compare log prices in terms of quality class, we find that the quality class significantly affects the price per cubic metre, while the differences between quality classes are not large. A significant price difference was found only between the $Q_1$ and $Q_4$ quality class logs (average quality sawlogs) (Table 5).

**Table 5.** Dependence of the price per cubic metre and the quality class of the spruce logs.

| Log Quality Class | n | Price per Unit Volume (EUR/m³) | Variance (EUR/m³) | Homogeneous Groups | Significantly Different from Class |
|---|---|---|---|---|---|
| $Q_1$ | 437 | 220.4 | 26.2 | X | 4 |
| $Q_2$ | 252 | 200.9 | 26.5 | XX | - |
| $Q_3$ | 119 | 184.2 | 28.5 | XX | - |
| $Q_4$ | 9 | 148.9 | 33.8 | X | 1 |

### 3.4. Artificial Neural Network Analysis

The neural network model showed that visual classification of logs by quality standards ($Q_1$–$Q_4$) was significantly related to the tested dendrochronological parameters of the logs as well as to acoustic data. Using ANN, we successfully classified logs by quality in 75.5% of cases (Table 6). Classification accuracy was best in quality class $Q_1$ (88.5%), followed by $Q_2$ (76.9%). This indicates that the dendrochronological and acoustic attributes of the highest-quality logs without structural anomalies define the relevant technological properties well enough. Sorting sawlogs ($Q_3$, $Q_4$) with ANN was worse and successful only in about two-thirds of cases. This suggests that structural anomalies known to be present in sawlogs [4,46–48] and increasing in quantity and dimension with decreasing quality class cannot be evaluated with the selected variables. It seems that, in the future, it would be useful to expand the list of variables to include the external geometric properties of logs and visually recognizable structural anomalies. Indeed, structural anomalies have been successfully used in several cases to classify wood by quality using ANN and image analysis [19,21,49].

**Table 6.** Confusion matrix output for the logs' quality data from the ANN model, with the numbers and percentages of classified spruce logs.

| | Predicted Quality Class | | | |
|---|---|---|---|---|
| **Actual Quality Class** | **$Q_1$** | **$Q_2$** | **$Q_3$** | **$Q_4$** |
| $Q_1$ | 23 (88.5%) | 3 (11.5%) | 0 (0.0%) | 0 (0.0%) |
| $Q_2$ | 2 (15.4%) | 10 (76.9%) | 1 (7.7%) | 0 (0.0%) |
| $Q_3$ | 0 (0.0%) | 4 (23.5%) | 12 (70.6%) | 1 (5.9%) |
| $Q_4$ | 0 (0.0%) | 0 (0.0%) | 1 (33.3%) | 2 (66.7%) |

With the optimized set of available dendrochronological and acoustic variables (Figure 6), we were less successful in classifying logs into price classes. Using ANN, we were only able to successfully classify slightly more than half (55.7%) of all spruce logs into price classes (Table 7). The ANN model showed relatively satisfactory classification accuracy only from $P_1$ to $P_3$ price classes, which accounted for 82.4% of all auctioned logs. Classification of ANN was unsuccessful for logs that achieved high ($P_4$–$P_6$) and extreme prices ($P_7$–$P_9$), mainly originating from the $Q_1$ quality class. As the price classes increased, so did the number of misclassifications, which were mostly false negatives.

**Table 7.** Confusion matrix output for the logs' price data per cubic metre from the ANN model, with numbers and percentages of classified spruce logs.

| | Predicted Price Class | | | | | | | | |
|---|---|---|---|---|---|---|---|---|---|
| **Actual Price Class** | **$P_1$** | **$P_2$** | **$P_3$** | **$P_4$** | **$P_5$** | **$P_6$** | **$P_7$** | **$P_8$** | **$P_9$** |
| $P_1$ | 8 (100%) | 1 (12.5%) | 0 (0.0%) | 0 (0.0%) | 0 (0.0%) | 0 (0.0%) | 0 (0.0%) | 0 (0.0%) | 0 (0.0%) |
| $P_2$ | 1 (4.0%) | 22 (88%) | 2 (8.0%) | 0 (0.0%) | 0 (0.0%) | 0 (0.0%) | 0 (0.0%) | 0 (0.0%) | 0 (0.0%) |
| $P_3$ | 0 (0.0%) | 1 (11.1%) | 7 (77.8%) | 1 (11.1%) | 0 (0.0%) | 0 (0.0%) | 0 (0.0%) | 0 (0.0%) | 0 (0.0%) |
| $P_4$ | 0 (0.0%) | 0 (0.0%) | 2 (40.0%) | 3 (60.0%) | 0 (0.0%) | 0 (0.0%) | 0 (0.0%) | 0 (0.0%) | 0 (0.0%) |
| $P_5$ | 0 (0.0%) | 0 (0.0%) | 0 (0.0%) | 2 (50.0%) | 2 (50.0%) | 0 (0.0%) | 0 (0.0%) | 0 (0.0%) | 0 (0.0%) |
| $P_6$ | 0 (0.0%) | 0 (0.0%) | 0 (0.0%) | 0 (0.0%) | 2 (66.7%) | 1 (33.3%) | 0 (0.0%) | 0 (0.0%) | 0 (0.0%) |
| $P_7$ | 0 (0.0%) | 0 (0.0%) | 0 (0.0%) | 0 (0.0%) | 1 (50.0%) | 0 (0.0%) | 1 (50.0%) | 0 (0.0%) | 0 (0.0%) |
| $P_8$ | 0 (0.0%) | 0 (0.0%) | 0 (0.0%) | 0 (0.0%) | 0 (0.0%) | 1 (50.0%) | 1 (50.0%) | 0 (0.0%) | 0 (0.0%) |
| $P_9$ | 0 (0.0%) | 0 (0.0%) | 0 (0.0%) | 0 (0.0%) | 0 (0.0%) | 0 (0.0%) | 1 (100.0%) | 0 (0.0%) | 0 (0.0%) |

Despite the fact that we defined price classes with a range of 100 EUR/m$^3$, we could not classify logs accurately with the tested set of variables. When we tested the classification of logs into narrower price classes using ANN, for example, with a range of 50 EUR/m$^3$, the accuracy of the classification was even worse. This indicates that the technological characteristics tested are not sufficient to predict the price of spruce logs, as opposed to classification by quality (Table 6). This is bad news for the owners (bidders) and the forest management advisors, because it shows that technological characteristics and log quality cannot be directly used to predict the price obtained at auction. The purchase price offered for the log also seems to depend on other factors which the log supplier has no direct influence on. According to other studies in this field, the price offered for logs in different markets is highly dependent on supply and demand, as well as on the specific needs and requirements of the buyer [2,50–52]. In isolated cases, probably because of the dimensional requirements of the products to be obtained from the logs and the high yield, buyers have offered extremely high purchase prices for logs simply because they were large in diameter or length but otherwise of average quality. If the quality is adequate, the price can be determined by the customer's individual requirements.

## 4. Conclusions

Examining Norway spruce (*Picea abies*) logs from the traditional national auction of valuable log assortments in Slovenj Gradec, Slovenia, we found that about half of the auctioned logs were of the highest quality ($Q_1$, $Q_2$) according to the standard and had a diameter of more than 60 cm. These logs were the most expensive, but the price difference compared to the thinner logs of the lower quality classes ($Q_3$, $Q_4$) was not large.

The highest priced logs where acoustic and tree-ring analyses were made were, on average, about 130 years old, had a tree-ring width of 2.65 mm, had a latewood proportion of 23%, and mainly originated from lower-elevation sites where Norway spruce is threatened by climate change.

The quality of the logs, determined by their external characteristics and geometry, was related to tree-ring and acoustic characteristics. The artificial neural network model (ANN) with feed-forward backpropagation using tree-ring data, longitudinal stress wave velocity, and damping showed that more than 75% of the logs could be accurately classified into quality classes. On the other hand, tree-ring data and acoustic characteristics could not adequately explain the price offered at auction. This suggests that the price offered most likely also depends on unidentified individual requirements and needs of the buyer. It is recommended that a larger experiment be conducted with wood from a greater number of sites to clarify the as yet unidentified factors that influence the prices of logs offered at auction.

Although the logs studied came from sites where spruce growth is potentially affected by climate change, the quality of the wood studied did not seem to be affected.

**Author Contributions:** Conceptualization and experiment design, A.S., K.Č. and K.N.; A.S. and K.N. performed the experiments; A.S. and K.N. analysed the data; validation and formal analysis, K.Č.; writing—original draft preparation, A.S.; writing—review and editing, K.Č. and K.N.; project administration, K.N.; funding acquisition, K.N and K.Č. All authors have read and agreed to the published version of the manuscript.

**Funding:** This work was supported by the project "Researchers-2.1-UL-BF-952011", contract no. C3330-19-952011; co-financed by the Ministry of Education, Science, and Sport of the Republic of Slovenia and the EU European Regional Development Fund and by the Slovenian Research Agency (ARRS) Programs P4-0015 (Wood and lignocellulosic composites) and P4-0430 (Forest timber chain and climate change: the transition to a circular bio-economy) and the research project CRP, V4-2016 LesGoBio supported by the Ministry of Agriculture, Forestry, and Food of the Republic of Slovenia (MKGP) and the Slovenian Research Agency (ARRS).

**Data Availability Statement:** The data presented in this study are not publicly available due to non-disclosure agreement.

**Acknowledgments:** We would like to thank Jože Jeromel and Klavdija Jeromel from the company Tiama Gozdarstvo d.o.o. (Podgorje, Slovenj Gradec, Slovenia) for their immense support and for enabling us to measure the logs at the auction field. We thank Paul Steed for editing the English.

**Conflicts of Interest:** The authors declare no conflict of interest. The funders had no role in the design of the study; in the collection, analyses, or interpretation of data; in the writing of the manuscript; or in the decision to publish the results.

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
