# Peer review of "Quality and Price of Spruce Logs, Determined Conventionally and by Dendrochronological and NDE Techniques"

_forests, doi:10.3390/f13050729_

Round 1
Reviewer 1 Report
Dear Authors,
It is a very carefully prepared and interesting manuscript expanding the knowledge of the Norway spruce wood. In my review, I will focus on only one element that should be improved (supplemented).
Chapter: “Materials and Methods”
Lines 85-89 and Table 1
Different countries have different evaluation (grading) systems for Roundwood, so it is not obvious what diameter is the basis for the classification of logs in this article.
There are different possibilities here:
- diameter at the smaller end of the log
- diameter of the larger end of the log
- diameter at the mid-section of the log
Please clarify (indicate) what diameter was used.
Yours sincerely
Reviewer
Author Response
Dear reviewer, thank you very much for the expert review and valuable comments. The missing information about how we determined the diameter of the logs was included in the sentence in line 86, which is now worded as follows:
The diameter of the logs, determined at the mid-section of the log, ranged from 43 to 84 cm, which was the basis for dividing the logs into diameter classes (n = 9) with a 5 cm interval: D1 (< 45 cm), D2 (45 - 49 cm), D3 (50 - 54 cm), D4 (55 - 59 cm), D5 (60 - 64 cm), D6 (65 - 69 cm), D7 (70 - 74 cm), D8 (75 - 79 cm), D9 (80 - 84 cm).
Reviewer 2 Report
Dear authors,
The manuscript provides a potentially interesting investigation of the quality and price of spruce logs, determined conventionally and by dendrochronological and NDE techniques. Authors examined selected 817 logs of Norway spruce with highest prices from a traditional auction in Slovenia, analyzed log dimensions and geometry, tree ring characteristics, quality grades according to the standard, properties measured by 11 NDE stress wave testing for modelling. Despite the complexity explained by influence of many factors on price of logs they showed that the artificial neural network model (ANN) with feed-forward backpropagation using tree-ring data and longitudinal stress wave velocity and damping can be used for classification into quality classes.
Minor points:
- It would be better to rewrite this sentence, because it is well-known that mechanical properties depend on bound water.
Ð .10, line 284-287
“Finally, it should be noted that the vibration resonant method, as used in this study, measures stress wave velocity and modulus of elasticity independently of moisture content, as these two values remain constant above the fibre saturation point [38,39].”
- For possible using suggested approach for other locations and wood species and comparative analyze, it would be better to add some data on range of wood density for this region provided in literature.
Author Response
Dear reviewer, thank you for the technical review and comment with the suggested changes in the article.
Comments:
- It would be better to rewrite this sentence, because it is well-known that mechanical properties depend on bound water.
Ð .10, line 284-287
“Finally, it should be noted that the vibration resonant method, as used in this study, measures stress wave velocity and modulus of elasticity independently of moisture content, as these two values remain constant above the fibre saturation point [38,39].”
Thank you for you remark. We have shortened the sentence and taken out the explanation of the mechanical properties, which are also related to density and indirectly to moisture. The sentence now explains more clearly the independence of the stress wave velocity from the moisture content above the FSP and the importance of the influencing factors explained earlier. The sentence is now as follows:
"Finally, it should be mentioned that the stress waves were analyzed in the green state of the logs, where their velocity is not influenced by the moisture content [38,39], but rather by the factors mentioned above."
- For possible using suggested approach for other locations and wood species and comparative analyze, it would be better to add some data on range of wood density for this region provided in literature.
Thank you very much for this important comment. We fully agree with you that it is necessary to include wood density in the study, which could further explain the classification of logs by quality. It was not specifically considered in this study, but otherwise varied with tree ring width, as we know from anatomy research on spruce. In particular, we intend to include density in the continuation of this study. We will include it as part of dendrochronological research by measuring blue intensity.